


# Gap-filling processes on GOCI-derived daily sea surface salinity product for Changjiang diluted water front in the East China Sea

Jisun Shin[1], Dae-Won Kim[2], So-Hyun Kim[3], Gi Seop Lee[4], Boo-Keun Kim[1,5], Young-Heon Jo[1,5*]

[1]BK21 School of Earth and Environmental Systems, Pusan National University, Busan, 46241, Korea
[2]Center for Climate Physics, Institute for Basic Science, Busan, 46241, Korea
[3]Department of Civil, Urban, Earth, and Environmental Engineering, Ulsan National Institute of Science and Technology, Ulsan, 44919, Korea
[4]Marine Bigdata AI Center, Korea Institute of Ocean Science and Technology, Busan, 49111, Korea
[5]Department of Oceanography and Marine Research Institute, Pusan National University, Busan, 46241, Korea

*Correspondence to*: Young-Heon Jo (joyoung@pusan.ac.kr)

**Abstract.** The spatial and temporal resolutions of contemporary microwave-based sea surface salinity (SSS) measurements are insufficient. Thus, we developed gap-free gridded daily SSS product with high spatial and temporal resolutions, which can provide information on short-term variability in the East China Sea (ECS), such as the front changes by Changjiang diluted

water (CDW). Specifically, we conducted gap-filling for daily SSS products based on the Geostationary Ocean Color Imager (GOCI) with a spatial resolution of 1 km (0.01°), using a machine learning approach during the summer season from 2015 to 2019. The comparison of the Soil Moisture Active Passive (SMAP), Copernicus Marine Environment Monitoring Service (CMEMS), and Hybrid Coordinate Ocean Model (HYCOM) SSS products with the GOCI-derived SSS over the entire SSS range showed that the SMAP SSS was highly consistent, whereas the HYCOM SSS was the least consistent. In the <31 psu

range, the SMAP SSS was still the most consistent with the GOCI-derived SSS ($R^2 = 0.46$; root mean squared error: RMSE = 2.41 psu); in the >31 psu range, the CMEMS and HYCOM SSS products showed similar levels of agreement with that of the SMAP SSS. We trained and tested three machine learning models—the find trees, boosted trees, and bagged trees models—using the daily GOCI-derived SSS as the ground truth, while including the three SSS products, environmental variables, and geographical data. We combined the three SSS products to construct input datasets for machine learning. Using the test dataset,

the bagged trees model showed the best results (mean $R^2 = 0.98$ and RMSE = 1.31 psu), and the models that used the SMAP SSS as input had the highest level. For the dataset in the >31 psu range, all models exhibited similarly reasonable performances (RMSE = 1.25–1.35 psu). The comparison with in situ SSS data, time series analysis, and the spatial SSS distribution derived from models showed that all models had proper CDW distributions with reasonable RMSE levels (0.91–1.56 psu). In addition, the CDW front derived from the model gap-free daily SSS product clearly demonstrated the daily oceanic mechanism during

summer season in the ECS at a detailed spatial scale. Notably, the CDW front in the horizontal direction, as captured by the Ieodo Ocean Research Station (I-ORS), moved approximately 3.04 km per day in 2016, which is very fast compared with the cases in other years. Our model yielded a gap-free gridded daily SSS product with reasonable accuracy and enabled the successful recognition of daily SSS fronts at the 1-km level, which was previously not possible with ocean color data. Such



successful application of machine learning models can further provide useful information on the long-term variation of daily
SSS in the ECS.

**1 Introduction**

Sea surface salinity (SSS)—the salinity of the ocean at its surface—affects the marine biogeochemical environments,
atmosphere–ocean interactions, and vertical ocean circulation (Dinnat et al., 2019; Durack et al., 2016). Gridded SSS dataset
is useful for research on climate change and its variability (Lyman and Johnson, 2014; Ciais et al., 2013; Domingues et al.,
2008; Bagnell and Devries, 2021). Because waters affected by river outflow and coastal regions are characterized by short-
term variability, gridded SSS products can provide useful information for monitoring SSS variations (Geiger et al., 2013; Chen
and Hu, 2017, Moon et al., 2019). The East China Sea (ECS)—a continental marginal sea in the western Pacific—receives
freshwater from the Changjiang (Yangtze) River, which is the fifth largest river based on discharge (Beardslev et al., 1985).
Changjiang River discharge (CRD) forms the Changjiang diluted water (CDW) by mixing with saline ambient waters and
causes seasonal and interannual changes in the ECS and Yellow Sea (YS) (Lie et al., 2003; Chen et al., 2008). In summer,
owing to the prevailing southerly wind and increasing CRD, the CDW extends eastward toward Jeju Island in Korea by
approximately 12–17 km per day and lasts approximately 1–2 months (Kim et al., 2009; Yamaguchi et al., 2012). The CDW
generally refers to seawater with salinity of no more than 31 psu. Low-salinity events caused by the CDW affects the
environment by altering the biological or physical properties of seawater, e.g., causing sea surface warming by impeding
vertical heat exchange (Chang and Isobe, 2003; Moon et al., 2019). Therefore, spatiotemporally continuous gridded SSS data
with a high spatial resolution and temporal resolution of at least a day are essential for monitoring the rapidly changing CDW
in the ECS.

Three approaches are mainly followed for SSS estimation : (1) Methods involving in situ observations, resulting in objective
analysis data products (Roemmich and Gilson, 2009; Cheng and Zhu, 2016; Lu et al., 2020); (2) data assimilation methods
using model-derived reanalysis data and combining numerical simulations with in situ observations (Forget et al., 2015;
Balmasede et al., 2013); and (3) methods involving satellite observations, i.e., passive microwave and ocean color products
(Reul et al., 2020; Chen and Hu, 2017; Wang and Deng, 2018; Kim et al., 2020; Kim et al., 2022a). First, the accuracy of in
situ observations defines how information is propagated from data-rich to data-sparse regions and is critically dependent on
data coverage and the reliability of spatial covariance (Von Schuckmann et al., 2014; Zhou et al., 2004). Hence, SSS products
obtained from in situ measurements involve the limitations regarding spatiotemporally continuous SSS monitoring over vast
areas. The Array for Real-time Geostrophic Oceanography (ARGO), which was established in the 2000s, provides in situ
measurements of various oceanographic parameters, including sea temperature and salinity, with a sparse array of $3° \times 3°$
(Dinnat et al., 2019; Vinogradova et al., 2019). ARGO monitors seas in various parts of the world. However, there are only
few in situ SSS observations from ARGO floats in the ECS (Kim et al., 2023b). Second, the model-derived reanalysis approach
relies on model simulations that use data assimilation schemes to constrain models based on various types of observations,



such as in situ and satellite data (Palmer et al., 2017; Cheng et al., 2020; Storto et al., 2019). Such products, particularly those below the ocean surface, may be significantly affected by model biases. Therefore, the accuracy of reanalysis products is lower than that of observational products when adopting a data assimilation approach in applications such as long-term climate change. The Hybrid Coordinate Ocean Model (HYCOM) and Copernicus Marine Environment Monitoring Service (CMEMS)

provide SSS fields in the ECS. Because these reanalysis data were generated and verified by mainly focusing on open-ocean conditions, the accuracy is low in waters with low, rapidly changing salinity levels, such as the ECS.

In contrast, satellite observations can resolve the limitations of in situ observations and reanalysis data. Three passive microwave radiometers with an L-band (1.4 GHz), including Aquarius (August 2011 to June 2015), Soil Moisture and Ocean Salinity (SMOS; since May 2010), and Soil Moisture Active Passive (SMAP; since April 2015), have been used for estimating

SSS. L-band sensors estimate SSS based on a dielectric constant model (Reul et al., 2020). Because SMOS does not provide SSS data in the ECS due to sensor errors, including land–sea contamination (LSC) and radio frequency interference (RFI) (Olmedo et al., 2018), only SMAP data are currently available. SMAP has been used to monitor SSS; however, uncertainties due to RFI and low sea surface temperature (SST) often lead to major errors, especially in river-dominated coastal waters, such as the ECS. To compensate for these limitations, Jang et al. (2021) attempted to improve the SMAP SSS in river-

dominated oceans using machine learning approaches. They used the SMAP SSS, Tb H-pol, Tb V-pol, Tb H/V, HYCOM SSS, SST, wind speed, and wave height as inputs and in situ data as the ground truth. Jang et al. (2022) produced a global SSS product by adding land fraction, distance from land, and precipitation data. However, the spatial (25–100 km) and temporal (5–7 days) resolutions of these data were too coarse to identify rapidly changing small mesoscale features in the ECS. In comparison, ocean color sensors, such as the Moderate Resolution Imaging Spectroradiometer (MODIS), Landsat series, and

Geostationary Ocean Color Imager (GOCI), can provide SSS products with high spatial and temporal resolutions (Wang and Deng, 2018; Chen and Hu, 2017). Specifically, GOCI, which operated from 2010 to 2021, had high spatial (0.5 km) and temporal (eight images per day) resolutions for monitoring short- and long-term SSS variations in the ECS. Several studies detected SSS variations using GOCI (Liu et al., 2017; Sun et al., 2019; Kim et al., 2020; Kim et al., 2021). Choi et al. (2021) analysed the variations in SSS, chlorophyll-a concentration, and SST when Typhoon Soulik passed over the study area and

revealed that decreasing salinity effects were strongly exhibited two days after the typhoon passed and then became weaker a week after the passage. Son and Choi (2022) elucidated the spatial and temporal CDW variations in the ECS through monthly GOCI-derived SSS maps for the 2011–2020 summer seasons. However, there has been a limit to the SSS estimation due to the wavelength band-associated calculations of the ocean color sensor, i.e., the nonlinear relationship between wavelength information of ocean color sensor data and SSS. In addition, only monthly SSS maps can be recognized, owing to severe cloud

contamination.

To overcome this problem, machine learning approaches have been used for SSS estimation. Kim et al. (2020) developed an SSS detection algorithm using a multilayer perceptron neural network (MPNN), which was applicable only for the summer of 2016. They used GOCI remote sensing reflectance ($R_{rs}$), SST, longitude, and latitude as inputs and SMAP data as the ground-truth. Kim et al. (2022a) performed a GOCI-II based SSS estimation in the ECS for the summer of 2021 using MPNN. They



provided the spatial distribution of low-salinity water near the southwestern Korean coasts at an hourly temporal resolution and a spatial resolution of 250 m, which was better than that of GOCI (500 m). For long-term SSS monitoring in the ECS, Kim et al. (2022b) trained the MPNN using Ocean Color Climate Change Initiative (OC-CCI) data and in situ data collected during the summer seasons of 1997–2021. They investigated the CDW front in the ECS using an SSS-estimated MPNN model. Monthly cumulative isohaline footprints revealed that the CDW propagates to the northeast and forms a longitudinally-oriented

ocean front. They mentioned that it is difficult to produce a monthly SSS distribution map because to frequent cloud cover, sun glint, and thick aerosols. In fact, because CDW progresses rapidly, SSS variations caused by CDW must be identified at a daily or finer temporal resolution. If gap-free daily SSS maps with high spatial resolutions can be obtained, the understanding of SSS variations in the ECS can be enhanced. Here, we performed gap filling for a GOCI-derived daily SSS product with a spatial resolution of 1 km (0.01°) using a machine learning approach. For this, we compared three SSS products, namely

SMAP, CMEMS, and HYCOM, in the ECS during the summers of 2015–2019. We then trained and tested three machine learning models, namely fine trees, boosted trees, and bagged trees, using the SSS product, environmental variables, and geographical data. Finally, we analyzed the CDW front in the ECS during summer using the gap-free GOCI-derived daily SSS product.

## 2 Materials

### 2.1 SSS and environmental data

Fig. 1 shows the study area (28.5–35°N, 119.5–129°E), including the ECS and YS. Table 1 presents a summary of the inputs and outputs used for model training and testing. All data were obtained according to the study area. The SMAP, HYCOM, and CMEMS SSS products were used as reference SSS data. Among passive microwave radiometers with L-bands, the SMAP product produced by the Jet Propulsion Laboratory (JPL) has a daily temporal resolution (eight-day running mean)

(https://podaac.jpl.nasa.gov/dataset). We used the version 5.0 SMAP-SSS level 3 product (SMAP_RSS_L3_SSS_SMI_8DAY-RUNNINGMEAN_V5), which has been available since March 27, 2015. SMAP went into safe mode and data collection was disrupted over 38 days from 17 June 2019 to 25 July 2019. The datasets are gridded to $0.25° \times 0.25°$. HYCOM is a data-assimilative hybrid isopycnal-sigma-pressure coordinate ocean model. Multiple datasets, including Argo data with in situ temperature and salinity (TS) profiles, satellite SST, and altimeter sea surface height (SSH)

anomalies, are used for HYCOM assimilation. We used Global Ocean Forecasting System (GOFS) 3.1 Global Analysis data (https://tds.hycom.org/thredds/catalog.html), with a temporal frequency of 3 h and a spatial resolution of $0.08° \times 0.08°$. We used the sea water salinity (SS) at a depth of 0.49 m of the CMEMS Global Ocean Physics Reanalysis data (GLOBAL_MULTIYEAR_PHY_001_030) (https://resources.marine.copernicus.eu/products). The GLORYS12V1 product is the CMEMS global ocean eddy-resolving reanalysis covering altimetry at $0.08° \times 0.08°$ and 50 standard levels. The

observations were assimilated using a reduced-order Kalman filter, and track altimeter data, satellite SST and in situ TS profiles were jointly assimilated.





## 2.2 Environmental and ground truth data

For the SST data, we used the Group for High Resolution SST (GHRSST) Level 4 Multi-scale Ultra-high Resolution (MUR) Global Foundation SST analysis version 4.1 data (https://podaac.jpl.nasa.gov/dataset/MUR-JPL-L4-GLOB-v4.1). The MUR

SST analysis is part of the NASA Making Earth System data records for Use in Research Environments (MEaSUREs) Program. The objective of creating the MUR SST was to develop a coherent and consistent daily SST map at the highest spatial resolution. The MUR SST has a spatial resolution of $0.01°$. For other environmental data, we used the SSH above the geoid, eastward sea water velocity (*uo*), and northward sea water velocity (*vo*) at a depth of 0.49 m of the GLORYS12V1 product. We used the eastward and northward components of 10-m wind datasets with $1/4°$ provided by the European Center for Medium-range

Weather Forecasts (ECMWF) Reanalysis v5 (ERA5). The wind data were converted to eastward wind stress (*wsu*) and northward wind stress (*wsv*) using an equation based on the air density, drag coefficient, and wind speed (Trenberth et al., 1990). Geographical data, such as longitude and latitude, used in the gridded data were matched to the gridded map with the scale of SST variable. For ground-truth data, we used the GOCI-derived SSS daily map of the ECS developed by Kim et al. (2021). They employed the MPNN approach using the hourly GOCI R$_{rs}$ product as input and SMAP SSS data as the ground

truth for 2015–2020.

## 2.3 In situ data

Fig. 1 shows the locations of the shipboard measurement data and the Ieodo Ocean Research Station (I-ORS) used to validate the model performance. We used serial oceanographic observation data provided by the National Institute of Fisheries Science (NIFS) serial oceanographic observation stations (http://www.nifs.go.kr/kodc/soo_list.kode). The station and observation

layers consist of 25 lines with 207 stations and 14 standard water column layers (0–500 m), respectively. Data are available from 1961 to present and are usually obtained six times a year, whereas in the case of the ECS, data are available four times a year. As shown in Fig. 1, we used SSS data with water level of 0 m obtained from the ECS (lines 315, 316, and 317), West sea (lines 311 and 312) and South sea (lines 203, 204, 205, 206, and 400). We obtained 861 SSS measurements at 103 observation points during the June–September period of 2015–2019.

The I-ORS salinity data were obtained from the Korea Institute of Ocean Science and Technology (KIOST) (https://kors.kiost.ac.kr/en/data/sub4.php). They are provided at depths of 3, 5, 8, 13, 18, 28, 34, and 40 m. The time interval is 10 min. We used salinity data at a depth of 3 m with daily averaging from June to September 2016 (122 days). The I-ORS is located at 125.18°E, 32.12°N. This station has an advantage in terms of low-salinity water monitoring because it is geographically located on the path of the CDW, extending from the Changjiang River to the waters of the Korean Peninsula.


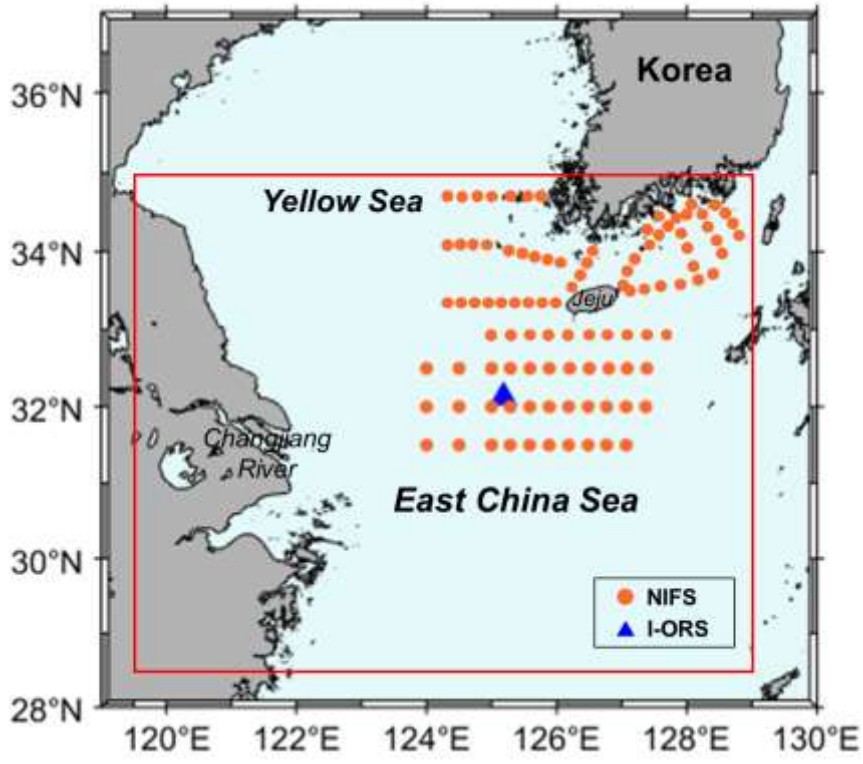

**Figure 1: The study area in the red solid box (28.5–35°N, 119.5–129°E) includes the East China Sea (ECS) and Yellow Sea (YS). Orange dots indicate the serial shipboard observation stations from the National Institute of Fisheries Science (NIFS), and blue triangle indicated the location of Ieodo Ocean Research Station (I-ORS). The two datasets were used for the model testing.**

**Table 1. Summary of inputs and outputs as the ground truth used for training and testing of the machine learning model. In situ SSS data for the model testing were provided by the NIFS and I-ORS.**

| Data type | Variable | Dataset | Data source | Horizontal resolution |
|---|---|---|---|---|
| Input | Sea surface salinity (SSS) | SMAP_RSS_L3_SSS_SMI_8 DAY-RUNNINGMEAN_V5 | SMAP | 0.25° × 0.25° |
| | | GOFS 3.1 GLBv0.08 | HYCOM | 0.08° × 0.08° |
| | | GLOBAL_REANALYSIS_P HY_001_030 | CMEMS | 0.08° × 0.08° |
| | Sea surface height (SSH) Eastward horizontal velocity (*uo*) Northward horizontal velocity (*vo*) | GLOBAL_REANALYSIS_P HY_001_030 | CMEMS | 0.08° × 0.08° |
| | Eastward component of 10 m wind (*wsu*) Northward component of 10 m wind (*wsv*) | ERA5 | ECMWF | 0.25° × 0.25° |
| | Sea surface temperature (SST) | MURSST | GHRSST Level 4 | 0.01° × 0.01° |
| | Geographically data (Longitude and Latitude) | - | - | 0.01° × 0.01° |
| Output | Daily SSS | GOCI-derived daily SSS (Kim et al., 2021) | GOCI | 0.01° × 0.01° |
| Validation | In situ SSS | In situ observations | NIFS and I-ORS | Point observation |



## 3 Methods

Fig. 2 shows a schematic representation of the generation of gap-free daily SSS product. In this study, we used a daily SSS
map at 3:00 UTC during the summer period (June–September) from 2015 to 2019 (610 days) estimated from GOCI $R_{rs}$. We
also obtained daily maps of other data for the same period. All data were sampled at 0.01°, which is the spatial resolution of
the SST level, to match the spatial resolution of the gridded maps. The SMAP, CMEMS, and HYCOM SSS products were
compared with the corresponding GOCI-derived daily SSS map as the ground truth through histograms, spatial distributions,
and scatter plots. In addition, the data were divided into below and above 31 psu, which is the standard for identifying the
CDW, and each of the two categories was evaluated for consistency with the corresponding GOCI-derived SSS. Thereafter,
the machine learning models were trained using a training dataset consisting of pixel pairs of GOCI-derived SSS and various
combinations of data, such as environmental factors and geographical data. We evaluated the quantitative performance of each
machine-learning model using a test dataset. After confirming the performances using in situ SSS, we investigated the time
series and spatial SSS distribution of each model. The optimal model was then selected. Finally, we analyzed the CDW front
in the ECS as estimated from the selected model. To determine the CDW front, we applied to a Savitzky–Golay filter with a
window size of four, which smooths according to a quadratic polynomial fitted over each window. This method is more
effective than other methods when the data vary rapidly. The SSS variations at the location of the I-ORS estimated by the
model were compared during the summers of 2015–2019, and the daily progress rate of the CDW was calculated using the
time-series diagram for the horizontal section at the latitude where the I-ORS is located.




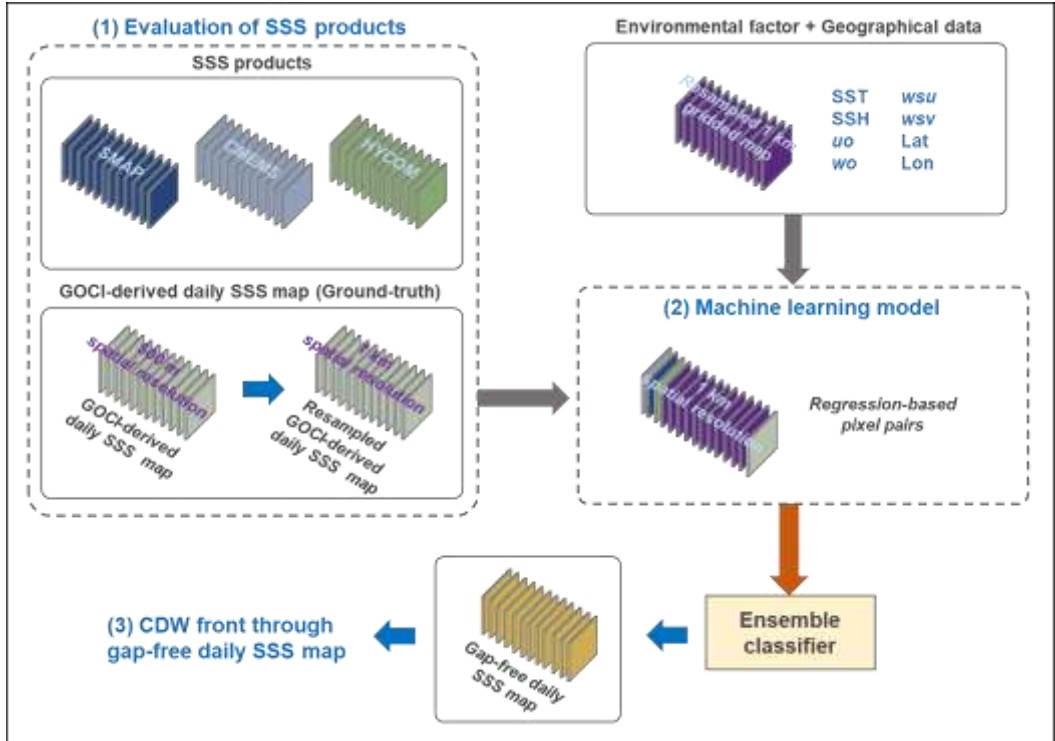

**Figure 2: Schematic diagram showing the processes that lead to the production of the gap-free Geostationary Ocean Color Imager (GOCI)-derived daily sea surface salinity map. We performed three steps: (1) We evaluated three SSS products in the ECS, including Soil Moisture Active Passive (SMAP), Copernicus Marine Environment Monitoring Service (CMEMS), and Hybrid Coordinate Ocean Model (HYCOM) SSS with the GOCI-derived SSS data, (2) machine learning models were trained and tested using the GOCI-derived SSS map and various combinations of data, including SSS products and environmental data, and (3) we identified the Changjiang diluted water (CDW) front generated from the gap-free daily SSS map.**

### 3.1 Machine learning models

Machine learning models were trained and tested using various input variable groups. Table 2 summarizes the composition of the input variables, number of pixel pairs, and training time for the models. To identify the extent to which the three SSS datasets affected the accuracy of the model, we created seven input variable groups: Three input groups (Models 1, 2, and 3) containing only one of the SMAP, CMEMS, and HYCOM SSS products; three input groups (Models 4, 5, and 6) containing combinations of two of the SSS datasets; and one input group containing all three SSS datasets (Model 7). Other data (SST, SSH, *uo*, *vo*, *wsu*, *wsv*, longitude, and latitude) were included in all the input groups. We matched the pixel pairs between the GOCI-derived daily SSS map and the corresponding factor maps. Because the SMAP SSS does not capture the coast due to its low spatial resolution, the input groups that included it had a small number of matched pixel pairs. In contrast, the input groups containing only the CMEMS and HYCOM SSS products exhibited 500,000 matched pixel pairs or more. For each model, the training and testing datasets were 80% and 20% of the matched pixel pairs, respectively. We then added as many as 10% of the matched pixel pairs to each input group. For example, the total number of matched pixel pairs in the input group 7 was 425,819, and the numbers of pixel pairs in the training and testing datasets were 340,656 and 85,163, respectively. By





adding a zero matrix of 10% for each pixel pair, the final numbers of pixel pairs for the training and testing datasets were 374,721 and 93,679, respectively. Using the seven input groups, we trained and tested three machine learning models, namely the (1) find trees, (2) boosted trees, and (3) bagged trees models. We used a fine regression tree with a minimum leaf size of four. Regression trees are easy to interpret, fast for fitting and prediction, and require low memory usage. Boosted trees are an

ensemble of regression trees that use a least-squares boosting algorithm. Compared to bagging, boosting algorithms use relatively little time or memory but might require more ensemble members. The minimum leaf size was set to eight, and the number of learners was 30 with a learning rate of 0.1 when the boosted trees model was trained. Bagged trees are bootstrap-aggregated ensembles of regression trees. They are often very accurate but can be slow and memory intensive for large datasets. The minimum leaf size and number of learners in the bagged trees model were the same as those in the booted trees model.

The characteristics of each model type were also revealed by the training results. As a result of confirming the training time for each model, it was found that the higher the number of pixel pairs, the longer the training time required for all models. In Model 2, which had the largest number of pixel pairs, the bagged trees required the longest training time of 185.97s. In terms of model type, the boosted trees model had the shortest training time, whereas the bagged trees model had the longest.

**Table 2. Composition of the input variables, number of pixel pairs, and training time required for each model. The three machine learning models, namely the fine trees, boosted trees, and bagged trees models, were trained and tested for estimating SSS from seven input variable groups.**

| Models | Input variables | | Number of pixel pairs | Training time (sec) | | |
|---|---|---|---|---|---|---|
| | SSS products | Other data | | Fine trees | Boosted trees | Bagged trees |
| Model 1 | SMAP | | 430,868 | 69.55 | 37.19 | 79.64 |
| Model 2 | CMEMS | | 567,946 | 150.43 | 93.97 | 185.97 |
| Model 3 | HYCOM | SST, SSH, *uo, vo, wsu, wsv,* lon, lat | 551,478 | 109.79 | 58.75 | 124.73 |
| Model 4 | SMAP+CMEMS | | 430,868 | 73.65 | 41.81 | 94.22 |
| Model 5 | SMAP+HYCOM | | 425,819 | 71.00 | 42.07 | 95.25 |
| Model 6 | CMEMS+HYCOM | | 551,376 | 112.82 | 63.12 | 135.95 |
| Model 7 | SMAP+CMEMS+HYCOM | | 425,819 | 73.50 | 45.90 | 97.51 |

### 3.2 Performance evaluation

Data comparison was performed using the coefficient of determination ($R^2$), root mean squared error (RMSE), mean squared error (MSE), and mean absolute error (MAE). The MSE is the square of the RMSE. The MAE is always positive and similar to the RMSE, but less sensitive to outliers. The formulae are defined as follows:

$$R^2 = \left( \frac{\sum_i (y_i - \bar{y})(x_i - \bar{x})}{\sqrt{\sum_i (y_i - \bar{y})^2 \sum_i (x_i - \bar{x})^2}} \right)^2, \tag{1}$$




$$RMSE = \sqrt{\frac{\sum_i (y_i - x_i)^2}{N}}, \tag{2}$$

$$MSE = \frac{1}{N}\sum_i (y_i - x_i)^2, \tag{3}$$

$\quad MAE = \frac{1}{N}\sum_i |y_i - x_i|, \tag{4}$

where $N$ is the number of pairs, $i$ represents an individual pair, $x$ and $y$ represent the GOCI-derived SSS and compared SSS, respectively, and $\bar{x}$ and $\bar{y}$ are the mean values of $x$ and $y$, respectively.

## 4 Results and discussions

### 240 4.1 Comparison of the SSS products with the GOCI-derived SSS

To confirm the characteristics of the SSS products in the study area, we examined the distribution trends of the SMAP, CMEMS, and HYCOM SSS products against that of the GOCI-derived SSS product (Fig. 3a). The distribution of the SMAP SSS product was the most similar to that of the GOCI-derived SSS product, with the median values of the SMAP and GOCI-derived SSS products being 31.04 and 30.86 psu, respectively. However, the SSS ranges of the CMEMS and HYCOM SSS

products, especially the one of the latter product, had high probabilities at values close to 35 psu and low probabilities in the range between 25 and 30 psu. For the HYCOM SSS product, there were no values below 20 psu. The median values of the CMEMS and HYCOM SSS products were 32.72 and 33.50 psu, respectively. The HYCOM SSS product showed the lowest degree of agreement with the GOCI-derived SSS product. Fig. 3b shows a clear shift of the CDW during summer in the GOCI-derived SSS map at 21 July 2017, 2:00 UTC, i.e., the date with the least masking (10.25%) due to cloud cover over the entire

study period (610 days). However, we were unable to confirm the movement patterns of the continuous CDW on a daily basis because of cloud masking in most SSS maps within the study period. Fig. 3c shows the spatial masking ratio of the GOCI-derived SSS maps with pixel units; the masking ratio was more than 95% around the Changjiang River estuary. The minimum masking ratio was 72%, and we estimated that all pixels in the study area cloud not provide SSS information for at least 439 of the 610 days. Fig. 3d–f shows the spatial distributions of the SMAP, CMEMS, and HYCOM SSS, respectively, acquired

on the day the GOCI-derived SSS map in Fig. 3b was acquired. In addition, we compared the distributions of the SSS data more clearly through scatter plots between the GOCI-derived SSS and the three SSS products (Figs. 3g–i). Consistent with the results in the scatter plot ($R^2 = 0.58$; RMSE = 1.97 psu), the SMAP SSS map showed the most similar distribution to that of the GOCI-derived SSS map; however, it did not reflect the daily SSS product because it was an 8-days average product. The





CDW pattern in the SMAP SSS was roughly consistent with that of the GOCI-derived SSS; however, in the western waters of
Jeju Island, the CDW pattern in the SMAP SSS did not appear like it did in the GOCI-derived SSS, thereby confirming that
the SMAP SSS was slightly overestimated compared with the GOCI-derived SSS in the scatter plot (Fig. 3g). In the case of
the CMEMS SSS, the CDW pattern in front of the Changjiang River estuary was similar to that of the GOCI-derived SSS, but
the CDW was distributed along the northern coast, and the high SSS area was expanded in the southern waters (Fig. 3e),
resulting in a form that deviated significantly from the 1:1 line ($R^2 = 0.27$; RMSE = 3.34 psu; MSE = 10.91), as shown in Fig.
2h. In contrast, the distribution of the HYCOM SSS had a large expansion of the high SSS area from south to north, and there
was no CDW pattern except at the front part of the Changjiang River, which had an extremely low SSS. In line with this, we
confirmed that the HYCOM SSS data were considerably overestimated compared to the GOCI-derived SSS data, i.e., $R^2 = 0.18$ and RMSE=3.68 psu, especially for the <31 psu case (Fig. 3i). Through the scatter plots, we confirmed that the degree of
agreement differed based on the 31-psu criterion. Table 3 shows the results of the calculation of the consistency with the
corresponding SSS products by dividing the GOCI SSS data based on the 31 psu criterion. Even in the <31 psu case, the SMAP
SSS still showed the best agreement with the GOCI-derived SSS ($R^2 = 0.46$; RMSE = 2.41 psu), whereas the HYCOM SSS
showed the worst agreement ($R^2 = 0.05$; RMSE = 4.86 psu). However, in the >31 psu case, the CMEMS and HYCOM SSS
products, with RMSE = 1.59 psu for the former and RMSE = 1.33 psu for the latter, showed as much agreement as that of the
SMAP SSS with RMSE = 1.20 psu. This was different from the results in the <31 psu case.

These results may be attributed to the following reasons. The characteristics of the reanalysis data may affect the SSS
estimation error in the ECS. Jang et al. (2022) compared the SMAP and HYCOM SSS products with in situ data from the
global ocean, including the Pacific, Tropical, Arctic Oceans, and the Amazon River Plume. They reported that the HYCOM
SSS in low-salinity regions (<32 psu), particularly in river-dominated coastal regions, exhibited high uncertainty. This may be
because the HYCOM SSS data were assimilated into the ARGO data, which are relatively limited in low-salinity regions. In
fact, the Argo database involves little data from our study area—the ECS. The HYCOM model uses SSS climatology and
monthly mean river discharge data and does not use satellite-derived SSS products capable of real-time observations. However,
these data are too coarse to reproduce the observed rapid changes in low-salinity water in narrow areas (Wallcraft et al., 2009;
Cummings and Smedstad, 2014; Wilson and Riser, 2016; Metzger et al., 2017). Similar to the HYCOM data, the CMEMS
data were also assimilated. Although reanalysis SSS data can be used to analyse daily changes in the waters in more detail
compared to the SMAP SSS data, reanalysis data have relatively low accuracy in regions with low, rapidly changing SSS.
Currently, the SMAP are the only satellite data that can provide a continuous spatial SSS distribution in the ECS, although it
is an eight-day average dataset and has a rough spatial resolution of 25 km. Hence, the SMAP data have been frequently used
as the ground-truth for SSS estimations using ocean color sensor data. Kim et al. (2021) used the SMAP SSS data as the
ground-truth in an SSS estimation model. The estimated SSS was reasonable, with $R^2 = 0.61$ and RMSE = 1.08 psu with
respect to in situ SSS. Because we considered the SSS data produced in Kim's model as the ground truth, the SMAP SSS
may—naturally—be the most consistent with the GOCI-derived SSS. In fact, L-band microwave sensor-retrieved SSS has
some limitations, such as errors due to anthropogenic RFI and LSC (Olmedo et al., 2019). In addition, the SMAP SSS has

significant uncertainty in the polar regions owing to the relatively low SST (Jang et al., 2022). This is because the sensitivity of emissivity to salinity decreases as SST decreases, thereby increasing the error in the SMAP SSS (Dinnat et al., 2019; Reul

et al., 2012). For this reason, the reanalysis data correlated more with the in situ data than the SMAP SSS in the high-salinity regions. However, the SMAP SSS in our study area showed a more reasonable degree of agreement with the in situ NIFS SSS compared to that of the reanalysis SSS; hence, the SMAP SSS data can be a good reference for monitoring the CDW in the ECS.

Figure 3: (a) Histogram of the GOCI-derived, SMAP, CMEMS, and HYCOM sea surface salinity (SSS) products in the study area. Spatial distributions of the (b) GOCI-derived, (d) SMAP, (e) CMEMS, and (f) HYCOM SSS data in the study area. (c) Spatial masking ratio of the GOCI-derived SSS maps based on the pixel unit. Scatter plots showing the consistency patterns between the GOCI-derived SSS and (g) SMAP, (h) CMEMS, and (i) HYCOM SSS data using the entire SSS range. Statistical indices: coefficient of determination ($R^2$), root mean squared error (RMSE), mean squared errors (MSE), and mean absolute error (MAE).



**Table 3: The $R^2$, RMSE, MSE, and MAE for the SMAP, CMEMS, and HYCOM SSS products with respect to the GOCI-derived SSS data according to the SSS range. The SSS data were divided into above and below 31 psu, which is the standard for defining CDW.**

| SSS range | Paired number | Salinity product | $R^2$ | RMSE (psu) | MSE | MAE |
|---|---|---|---|---|---|---|
| < 31 psu | 225,203 | SMAP | 0.41 | 2.46 | 6.05 | 1.68 |
| | | CMEMS | 0.12 | 4.29 | 18.37 | 3.42 |
| | | HYCOM | 0.05 | 4.86 | 23.61 | 3.95 |
| > 31 psu | 200,616 | SMAP | 0.34 | 1.20 | 1.45 | 0.82 |
| | | CMEMS | 0.23 | 1.59 | 2.54 | 1.04 |
| | | HYCOM | 0.20 | 1.33 | 1.77 | 1.07 |


## 4.2 Performance of the SSS models

### 4.2.1 Quantitative evaluation

Table 4 summarizes the $R^2$, RMSE, MSE, and MAE values of the estimated SSS with respect to the GOCI-derived SSS for the model using the test dataset. Based on the statistical indices, the bagged trees model showed the best results (mean $R^2$ = 0.98 and RMSE = 1.31 psu), while the boosted trees model had the lowest accuracy (mean $R^2$ = 0.94 and RMSE = 2.14 psu). The MSE and MAE values suggested the same. The mean MSE and MAE of the boosted trees model were 2.63 and 2.23 times higher than those of the bagged trees model, respectively. The find trees model performed well, with a mean RMSE of 1.57 psu. Model 1 with the bagged trees model and only the SMAP SSS as input had the highest level ($R^2$ = 0.98 and RMSE = 1.16 psu). Models 4, 5, and 7 with the bagged trees models and the SMAP SSS as input also had statistical results similar to that of Model 1, with RMSE values of 1.17–1.19 psu. Model 2 with the boosted trees model and only the CMEMS SSS as input showed the worst results ($R^2$ = 0.93, RMSE = 2.41 psu, MSE = 5.82, and MAE = 1.78). However, the RMSE values were reasonable for the models with the boosted trees model and the SMAP SSS as input (Model 1, 4, 5 and 7). Notably, the RMSE values of Models 2, 3, and 6 with the bagged trees model and without the SMAP SSS as input showed relatively reasonable levels compared to the models that used for the SMAP SSS as input. This indicates that the bagged trees model overcomes the inconsistencies of the CMEMS and HYCOM SSS with respect to the GOCI-derived SSS compared to the fine trees and boosted trees models. Our results are similar to those of previous studies. Jang et al. (2022) improved the accuracy of the SMAP SSS for the global ocean using environmental data, the SMAP and HYCOM SSS, and various machine learning approaches. They reported that, among the models, ensemble tree-based machine learning methods such as random forest (RF), extreme gradient boosting (XGBoost), light gradient boosting model (LGB), and gradient-boosted regression trees (GBRT), showed quantitatively good performances. Shin et al. (2022) evaluated machine learning models with various types of ensemble methods, such as bagged trees, boosted trees, subspace discriminant, subspace k-nearest neighbor (KNN), and random undersampling boosting (RUSBoost), to estimate *Sargassum* distribution through environmental variables. They found that



model accuracy varied depending on the learner type and that the bagged tree model showed the best performance, especially when the learner type was a decision tree.

The following trials and errors occurred during the process of generating the dataset for the machine learning modeling. Because to the difference in spatial resolution among the data, the pixels masked at zero were different; therefore, they were adopted as pixel pairs only if all input values were available. This is because, when the zero value of a masked pixel is added as an input value, the accuracy of the estimated SSS value decreases. The model was trained using zero-free data, thereby not properly recognizing the actual mask pixels, resulting in a specific value of pixels in the masked area. Therefore, when

generating the training dataset, we added as many as 10% of the matched pixel pairs for each model. To recognize the impact of geographic factors on model performance, we trained the bagged trees model while excluding the latitude and longitude from the input data of Model 1. As a result, the RMSE, MSE and MAE values increased by 12.55%, 26.68%, and 12.99% compared to those of Model 1 with geographic factors, respectively. Spatially, the CDW pattern estimated from the model was more dispersed; therefore, the tendency of the movement pattern was not clear. The results of some previous studies are

consistent with these results. Shin et al. (2022) reported that the model trained with geographic factors as input variables was more accurate than the model without geographic factors, and that *Sargassum* distribution in the ECS estimated from the model was less spread and more reasonable than those from other models. Kim et al. (2023a) selected physically related variables and geographic factors as inputs to estimate the subsurface salinity using a convolutional neural network (CNN) model. They found that the model without geographic information was less accurate than the model with geographic information.

Geographic information is important for the movement of the CDW in the ECS. The Changjiang River, located in the southwestern part of the study area, is a major source of freshwater, and the CDW produced from this location gradually moves northeast. Therefore, a model with good performance model is possible only when both geographical and environmental factors that can affect the SSS variations are considered.




Earth System
Science
Data

**Table 4. Statistical results of the R², RMSE, MSE, and MAE values between the SSS products estimated from the seven models and the GOCI-derived SSS using the test dataset. The models were divided according to the SSS data used in input, and each training dataset was trained with the three machine learning models, namely fine trees, boosted trees, and bagged trees models.**

| SSS model | | $R^2$ | RMSE (psu) | MSE | MAE |
|---|---|---|---|---|---|
| **Model 1 (SMAP)** | Fine trees | 0.98 | 1.38 | 1.92 | 0.78 |
| | Boosted trees | 0.95 | 1.95 | 3.80 | 1.55 |
| | Bagged trees | 0.98 | 1.16 | 1.36 | 0.67 |
| **Model 2 (CMEMS)** | Fine trees | 0.96 | 1.88 | 3.55 | 0.95 |
| | Boosted trees | 0.93 | 2.41 | 5.82 | 1.78 |
| | Bagged trees | 0.97 | 1.56 | 2.43 | 0.82 |
| **Model 3 (HYCOM)** | Fine trees | 0.96 | 1.77 | 3.15 | 0.92 |
| | Boosted trees | 0.94 | 2.37 | 5.62 | 1.76 |
| | Bagged trees | 0.97 | 1.48 | 2.19 | 0.79 |
| **Model 4 (SMAP+CMEMS)** | Fine trees | 0.98 | 1.41 | 1.98 | 0.78 |
| | Boosted trees | 0.95 | 1.95 | 3.81 | 1.54 |
| | Bagged trees | 0.98 | 1.19 | 1.41 | 0.67 |
| **Model 5 (SMAP+HYCOM)** | Fine trees | 0.98 | 1.38 | 1.92 | 0.78 |
| | Boosted trees | 0.95 | 1.95 | 3.82 | 1.54 |
| | Bagged trees | 0.98 | 1.17 | 1.36 | 0.66 |
| **Model 6 (CMEMS+HYCOM)** | Fine trees | 0.96 | 1.77 | 3.13 | 0.91 |
| | Boosted trees | 0.94 | 2.38 | 5.66 | 1.76 |
| | Bagged trees | 0.98 | 1.47 | 2.16 | 0.78 |
| **Model 7 (SMAP+CMEMS+HYCOM)** | Fine trees | 0.98 | 1.38 | 1.92 | 0.77 |
| | Boosted trees | 0.95 | 1.96 | 3.83 | 1.54 |
| | Bagged trees | 0.98 | 1.18 | 1.38 | 0.66 |

To evaluate the performance of the bagged trees models, we validated the estimated SSS from the models using in situ NIFS and I-ORS SSS (Fig. 4). In the case of in situ NIFS (Figs. 4a–g), Model 1 with the bagged trees model had the best performance with $R^2$ = 0.65 and RMSE = 1.43 psu while Model 3 had the worst performance with $R^2$ = 0.59 and RMSE = 1.56 psu. Consistent with the results of using the test dataset, the models with the SMAP SSS as input (RMSE = 1.43–1.49 psu) performed slightly better than the models without (RMSE = 1.53–1.56 psu). As shown in Fig. 1, in situ data were acquired from within the area of the CDW, with a minimum SSS of 21.97 psu. Quite a few data were acquired from the northern location, which was unaffected by the CDW; therefore, the maximum SSS was 34.04 psu. However, of the 861 in situ data, the ones in the >31 psu range accounted for 73.17% of the total. Therefore, we evaluated model performance by dividing the data based on the 31 psu criterion (Table 5). When using in situ data in the >31 psu range, the mean RMSE was 5.36% higher than when using the entire dataset, and the performances of all models were slightly worse. However, when in situ data in the <31 psu range were used, the mean RMSE decreased by 12.32%. In particular, the RMSE of Model 2 (with only the CMEMS SSS as input) decreased the most, by 14.68%, compared with when all data were used. Notably, model performance with data in the < 31 psu range was similarly reasonable in all seven models (RMSE = 1.25–1.35 psu). The bagged trees model successfully solved the nonlinear relationships between the three SSS datasets and the GOCI-derived SSS dataset. In addition, the



performance of the model was evaluated using in situ I-ORS data from 2016, when the expansion scale of the CDW was quite large and fast (Figs. 4h–n). From June to September 2016, 16 out of 122 days were missing, and a total of 106 matching data points were used to evaluate the performance of the models. Within this period, the minimum SSS was 26.62 psu, and unlike
the in situ NIFS data, data in the <31 psu range accounted for 89% of the total data, with a maximum SSS of 32.02 psu. The RMSE range of all models was 0.91–1.02 psu, with good performance at a low salinity range. When confirming the consistency between the in situ I-ORS dataset and the three SSS datasets, the RMSE values of the SMAP, CMEMS, and HYCOM SSS were 1.46, 3.06, and 3.25 psu, respectively. The RMSE of Model 5, which had the worst performance among the models. These results were quite different in the case of the in situ NIFS data. As shown in table 3, the three SSS datasets showed high
accuracies in the >31 psu range; therefore, using the in situ NIFS dataset resulted in low RMSE values. However, the in situ I-ORS data have a high RMSE because most of the SSS data are in the <31 psu range. In addition, the spatial resolution of the reanalysis data may be too rough to capture the daily variations of the CDW moving 12–17 km per day. In contrast, the SSS map with 1km spatial resolution estimated by the models can be good at quickly catching the daily CDW variations. In summary, all bagged trees models trained with a combination of various input variables could estimate the SSS of the CDW
in the ECS on a daily basis with RMSE values of less than 1 psu, i.e., higher than that of the SMAP SSS.



**Figure 4:** Scatter plots of the in situ (a–g) NIFS data and (h–n) I-ORS data *versus* the SSS values estimated from Models 1–7 for the bagged trees model. N is the number of matches between the in situ SSS and estimated SSS maps. As shown in Fig. 1, in situ data were acquired from close to the coast of the Korean Peninsula; therefore, the SMAP SSS does not provide data around the coast. For this reason, the SSS map estimated from models using the SMAP SSS as input had fewer matched data than those of other models. Since I-ORS is located in the center of the study area, in situ data that matched data on the SSS map estimated from models were the same for all models.

**Table 5: The R², RMSE, MSE, and MAE values of the models with respect to the GOCI-derived SSS data based on the SSS range.**

| SSS range | Model | $R^2$ | RMSE (psu) | MSE | MAE | Matched number |
|---|---|---|---|---|---|---|
| < 31 psu | Model 1 | 0.45 | 1.25 | 1.56 | 0.89 | 183 |
| | Model 2 | 0.41 | 1.30 | 1.69 | 0.92 | 229 |
| | Model 3 | 0.42 | 1.35 | 1.82 | 0.96 | 207 |
| | Model 4 | 0.41 | 1.30 | 1.68 | 0.94 | 183 |
| | Model 5 | 0.40 | 1.35 | 1.83 | 0.94 | 166 |
| | Model 6 | 0.46 | 1.29 | 1.67 | 0.94 | 207 |
| | Model 7 | 0.43 | 1.32 | 1.74 | 0.94 | 166 |
| > 31 psu | Model 1 | 0.28 | 1.54 | 2.36 | 1.37 | 295 |
| | Model 2 | 0.25 | 1.60 | 2.56 | 1.44 | 612 |
| | Model 3 | 0.25 | 1.63 | 2.65 | 1.47 | 588 |
| | Model 4 | 0.31 | 1.53 | 2.35 | 1.38 | 295 |
| | Model 5 | 0.26 | 1.56 | 2.43 | 1.39 | 291 |
| | Model 6 | 0.27 | 1.60 | 2.57 | 1.45 | 588 |
| | Model 7 | 0.26 | 1.54 | 2.39 | 1.37 | 291 |

### 4.2.2 Time series and spatial distribution of SSS map

We compared the spatial distributions of the SSS maps to qualitatively evaluate the models. Fig. 5 shows the time series variations of the model-based SSS, GOCI-derived SSS, and in situ I-ORS SSS during the summer period, i.e., from 1 June
2016 to 30 September 2016. Out of a total of 122 days, GOCI-derived SSS data were obtained at the I-ORS location in 48 days, while 60.66% of the SSS data over four months were not observable due to cloud cover. The in situ I-ORS data were missing 13.11% of the data during the same period. However, the SSS data estimated by the models spanned over the entire period and did not include missing data, and simulated daily variation of the in situ data was better than that of the GOCI-derived SSS. Fig. 6a shows the GOCI-derived SSS map on 27 July 2016, in which the in situ I-ORS SSS value is the lowest,
as shown in Fig. 5. SSS data at the location of the I-ORS (red triangle) existed in the GOCI-derived SSS map; however, most parts of the study area were masked by clouds, making it difficult to recognize the CDW pattern. In contrast, Figs. 6b–h show the SSS maps estimated by Models 1–7, respectively, for the same date as that of the GOCI-derived SSS map. Unlike the GOCI-derived SSS map, all the SSS maps estimated by the models provided gap-free SSS distributions and clearly showed that the CDW extended from the Changjiang River estuary to the coast of Jeju Island during summer. However, we confirmed
that Models 1, 4, 5, and 7 (Figs. 6b, e, f, and h, respectively), which included the SMAP SSS as input, are masked in coastal areas, and some of the spatial patterns of the CDW appear in steps because of the spatial resolution of the SMAP (25 km). In contrast, Models 2, 3, and 6 (Figs. 6c, d, and g, respectively), which included the CMEMS and HYCOM SSS as inputs, did not mask the coastal area and provided coastal SSS information regarding the CDW spreading from the front of the Changjiang River estuary. Overall, the CDW patterns in the SSS maps estimated by Models 3 and 6 (Figs. 6d and g) using the HYCOM
SSS as input data were similar to those of the other models; however, the CDW distribution tended to be divided around 124° E. Model 2, which included only the CMEMS SSS as input, showed the appropriate CDW distribution and patter. Through

quantitative and qualitative evaluation of the models, we selected Model 1 (only the SMAP SSS as input) and Model 2 (only the CMEMS SSS as input) for the CDW front analysis in the ECS, while considering the simplicity of the input data. The prime SMAP mission was completed in the summer of 2018, having acquired a wide range of scientific data for three years.
Since then, SMAP has been approved for an extended phase operation until 2023. However, the SMAP operation will soon be terminated; therefore, an alternative to the SMAP SSS data is necessary. By quantitative and qualitative model evaluation using a combination of SSS products, we confirmed the usefulness of additional CMEMS SSS data, combined with the SMAP SSS data, to generate a gap-free GOCI-derived SSS map.

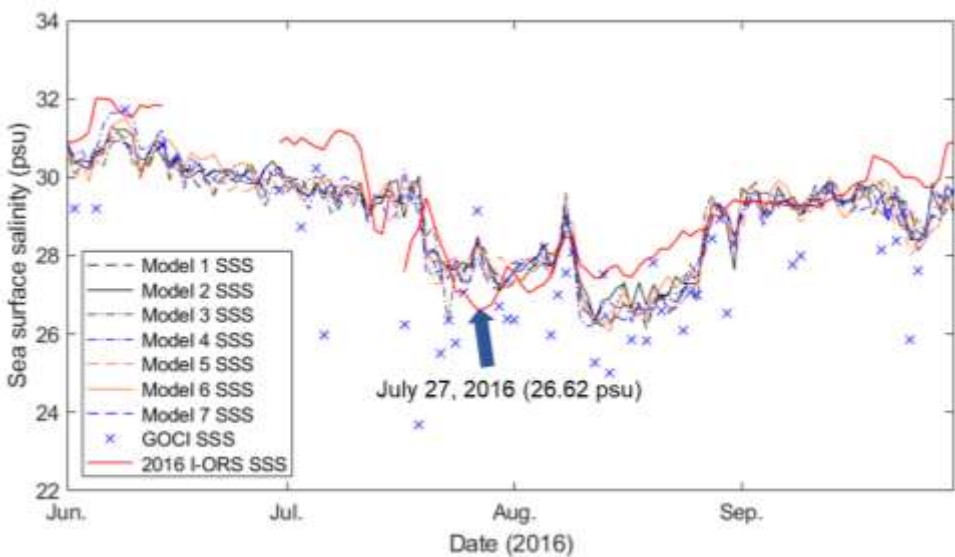


**Figure 5: (a) SSS time series estimated from seven models with bagged trees and the GOCI-derived and in situ I-ORS SSS data over 122 days from 1 June 2016 to 30 September 2016.**





**Figure 6: (a) GOCI-derived SSS map on 27 July 2016. (b)–(h) SSS maps estimated from seven models with the bagged trees. The**
**estimated SSS maps were generated from input data on the same day as the GOCI-derived SSS map. The red triangle represents**
**the I-ORS location.**

### 4.3 CDW front based on gap-free daily SSS

Fig. 7a shows the SSS time series estimated from Model 1 and 2 with bagged trees at the I-ORS location during the summers
of 2015–2019. We identified the three phases according the CDW variations during summer: (Phase I) beginning phase (early
June), (Phase II) development phase (end of July), and (Phase III) recovery phase (end of August). The in situ I-ORS SSS
generally began to fall under the influence of the CDW in June (Phase I), declined from July to August (Phase II), and then
increased in September (Phase III). This happened in 2015, 2016, and 2019, and the difference between the maximum and
minimum SSS was approximately 3 psu. However, 2016 and 2018 exhibited different trends. In 2016, the SSS change in Phase
I was similar to those in other years, whereas Phase II showed a sharp SSS decline, contrary to the cases in other years. At the
end of August, Phase III showed a sharp SSS increase and SSS recovered to a level similar to those in other years. In contrast,





in 2018, Phases I and III showed patterns similar to those in other years, whereas Phase II showed a slight SSS increase, contrary to the case in 2016. To determine the direction and velocity of the CDW front movement in the summers of 2015–2019, we plotted the time series in the cross-sectional direction (A–A´ in Fig. 8a). In early June, the CDW front was similarly located near 126°E in all years, and the 29 isohaline appeared near 124°E. In 2015 and 2019, the CDW front expanded to

127°E and gradually moved east until September. In 2017 and 2018, the CDW front did not reach 127°E until September, thereby repeating the trend of heading east and then retreating to the west. In an unusual case in 2016, we confirmed that the CDW front extended to 128°E on 1 August, 62 days after 1 June, moving approximately 3.04 km per day (188.29 km/62 days). Regarding the 29 isohaline, in 2017 and 2018, similar to June, it rarely moved east; in particular, in 2018, the tendency to move east was low, resulting in the lowest CDW expansion during the study period. In 2015 and 2019, the 29 isohaline developed

at 125°E in August and gradually retreated westward. In 2016, the 29 isohaline extended to 127°E from early June to early August, moving at a speed of approximately 4.79 km per day (282.42 km/59 days). This was faster than the CDW front, which lasted for one month in August and then gradually retreated in September. The 27 isohaline stayed around 123°E from early June to the end of September in all years, except 2016. In 2016, a partial 27 isohaline extended to 126°E, confirming that a fairly low-salinity environment persisted during the summer season of 2016.

Focusing on 2016 and 2018, which showed unusual SSS fluctuations different from those in other years, we continuously (i.e., on a daily basis) identified the CDW front (<31 psu) by phase (Fig. 8). The SSS spatial distributions were estimated by Model 1 and were applied with the 29, 31, and 33 psu isohalines for the CDW front. Consistent with the SSS time series in Fig. 7, the CDW front was close to the I-ORS during Phase I in early June (5–8 June) in both 2016 and 2018 (Figs. 3a and 8d, respectively). This indicates that before June, the CDW front had already advanced considerably eastward in the ECS and began to enter the

CDW boundary of the <31 psu range in Phase I. However, the CDW front variation patterns in Phases II and III were different. On 19–20 July 2016, during Phase I, the I-ORS location entered the 29 psu boundary, and the CDW front gradually expanded southeast. Conversely, in 2018, Phase I stayed at the 29 psu boundary at the same time and escaped, and the CDW front remained similar without significant changes. While Phase III in 2016 remained at the boundary of 29 psu boundary and gradually escaped, Phase III in 2018 exhibited a spatial CDW front pattern similar to that of Phase I, and the SSS level had

already recovered.

Our results are consistent with those of previous studies on the CDW in the ECS. Moon et al. (2019) recognized that ocean salinity in 2016 was exceptionally low and investigated the contribution of low salinity to sea surface warming in the ECS during the summer of 2016. Through observations, they revealed that a large amount of freshwater in 2016 originated from the Changjiang River. Son and Choi (2022) presented maps that applied various SSS algorithms to GOCI and noted that surface

water in the summer of 2016 was loaded with fresh water owing to increased Changjiang River discharge (CRD). In addition, cross- and along-shelf exports of the CDW from the Changjiang River mouth manifested as patches, and salinities below 25 psu were observed along the Changjiang River estuary. Kim et al. (2023a) estimated the CDW volume in the ECS by combining a subsurface salinity map with the SMAP SSS. The CDW volume was highest in 2016, whereas in 2018, it reached a minimum during summer. They found that the CDW volumes were relatively low from May to early June and increased



from June to August, showing a seasonal trend. This may be because the conditions in 2016 and 2018 were different, owing to the amount of CRD, precipitation, El Niño–Southern Oscillation (ENSO), typhoons, and wind. The primary factor controlling the scale of the CDW is the amount of CRD. Kim et al. (2023) reported that the amount of CRD measured at the Datong Station was highest in 2016 and lowest in 2018. They investigated the relationship between CDW volume and CRD and found that the CDW volume peak appeared with a time lag of about 34±15 days after an increase in CRD, and that 2016

had the largest CDW and 2018 was the smallest. In 2016, a strong El Niño event led to a noticeable increase in CRD compared to other years (Kim et al., 2023a). ENSO can increase the CRD in the ECS through the increased precipitation during El Niño events (Park et al., 2011; Wu et al., 2023). In addition, no typhoons crossed the ECS in 2016, indicating that no vertical mixing was caused by typhoons. Strong vertical mixing caused by the passage of a typhoon hinders the CDW expansion. In contrast, the La Niña event in 2018 led to a low CRD and typhoons crossed the ECS. These differences may change the CDW pattern

annually.

**Figure 7:** (a) SSS time series estimated by Model 1 (2015, 2016, 2017, and 2018) and Model 2 (2019) with bagged trees at the I-ORS location during the summer of 2015–2019. (Phase I) Beginning phase, (Phase II) development phase, and (Phase III) recovery phase of the CDW. In 2019, it was not possible to estimate the SSS with Model 1 because the SMAP SSS was not provided due to safe mode; therefore, Model 2 was used instead. (b)–(f) Time series of the 122–128°E horizontal transect (A–A´ in Fig. 8a) during summer in each year. The plots were applied for the 27, 29, and 31 psu isohalines. The cross section in A–A´ is located at 32.12°N.







**Figure 8: SSS spatial distribution in the ECS with the 29, 31, and 33 psu isohalines in 2016 and 2018. (a)–(f) Maps corresponding to the stages in Figs. 7a, b, c, respectively. Beginning phase (Phase I): panels (a) and (c); development phase (Phase II): panels (b) and (d); recovery stage (Phase III): panels (c) and (f). The red triangle represents the I-ORS location (32.12°N, 125.18°E).**

**5 Data availability**

The gridded gap-free SSS dataset at $0.01° \times 0.01°$ spatial resolution during summer period (June–September) from 2015–2019
is stored at the Korea Institute of Ocean Science & Technology (https://10.22808/DATA-2023-2, Shin et al., 2023).

**6 Summary**

To date, the SMAP satellite data and CMEMS and HYCOM reanalysis data are the gap-free gridded SSS products that can be
used in the ECS. We found that the reanalysis data showed fair accuracy with respect to the GOCI-derived SSS in the >31 psu
range. The worst agreement was found in the <31 psu range, which is the most oceanic environment in the ECS during summer.
Hence, the reanalysis SSS was not suitable for gap-filling in the GOCI-derived SSS. Because the SMAP SSS dataset is an
eight-day average dataset, the accuracy of the daily analysis was poor and had a fairly rough spatial resolution of 25 km;
however, to date, it is the only dataset that can grasp the gap-free daily spatial SSS distribution with fair accuracy in the <31
psu range. Nevertheless, we overcame the limitations of these datasets and succeeded in producing a gap-free gridded SSS
product with reasonable accuracy and a spatial resolution of 1 km using a machine learning approach and the corresponding
variable of SSS estimation. Our study facilitates the identification of the distribution and movement patterns of the CDW front
in the ECS on a daily basis during summer, thereby further advancing our understanding and monitoring of long-term SSS
variations.

**Author contributions**

JS developed the related model and evaluated, and wrote the manuscript. YH conceptualized and supervised this study. DW
and SH participated in data processing. YH, BK, and SH revised and reviewed the manuscript.

**Acknowledgements**

The serial oceanographic observation and I-ORS data was provided by the NIFS and the KOIST, respectively. The authors
would like to thank the staff of the data management.

**Financial support**

This work was supported by the National Research Foundation of Korea (NRF) grant funded by the Korea government (MSIT)
(RS-2023-00280650; RS-2023-00274972).



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
