# Peer review of "Gap-filling techniques applied to GOCI-derived daily sea surface salinity product for Changjiang diluted water front in the East China Sea"

_Earth System Science Data, 2023_

## Author Comment (AC1)

**[RC1]**

Dear Editor,

I have carefully reviewed the manuscript entitled *Gap-filling processes on GOCI-derived daily sea surface salinity product for Changjiang diluted water front in the East China Sea*. In this paper, the authors employ decision trees to derive a sea surface salinity (SSS) dataset for the Yellow Sea and East China Sea. Specifically, the input data comprises sea surface salinity (SSS) from three data products: GLORYS12, SMAP, and HYCOM, accompanied by other data such as sea surface temperature (SST), sea surface height (SSH), velocity, and wind stress curl. Notably, the "ground-truth" data used by the authors is the GOCI-derived SSS data from Kim et al. (2021), which was derived from SMAP data. The methods employed are three decision tree-based algorithms: fine trees, boosted trees, and bagged trees. The authors first compare the three SSS data products with the GOCI-derived SSS, then test different input combinations to determine the optimal inputs and algorithms. They demonstrate that the model with SMAP as input yields the highest accuracy, and the bagged trees perform best. However, models using the other two data products with bagged trees also produce satisfactory results. Comparisons between the SSS derived from the different models and SSS observations all yield coefficients of determination ($R^2$) greater than 0.6. Ultimately, the authors decide to utilize the models with SMAP and CMEMS GLORYS data inputs as the "final results" for further analysis. In the last section, the authors use the SSS derived from Model 1 (SMAP input) and Model 2 (GLORYS input) to describe the evolution of the Changjiang Diluted Water in the East China Sea during the period of 2015-2019.

Despite the technical proficiency exhibited in applying fine trees, boosted trees, and bagged trees algorithms, and the methodological rigor in comparing and selecting data inputs and algorithms, I have significant reservations that compel me to **recommend rejection of the manuscript for the following reasons**:

**Ground-Truth Data Appropriateness:** The designation of GOCI-derived SSS as "ground-truth" is fundamentally flawed. "ground-truth" refers to direct, in-situ measurements used to validate remote sensing products. GOCI-derived SSS from Kim et al. (2021), being a remotely sensed product itself, cannot serve as ground-truth, undermining the study's validation framework.

➔ Thank you for the comment. As you know, it is very difficult to produce daily gap-free SSS data using optical data with a spatial resolution of about 1km. Therefore, we performed the task of filling the gap in the data using the data that can be the basis of the gap-free product. As you pointed out, we fully agree with the reviewer and it is not appropriate to express the data by Kim et al. (2021) as "ground-truth". We used Kim et al. (2021) data as the output data for the model's training, and I think there was a misunderstanding by expressing it as ground-truth data. Therefore, we have changed the terms "ground-truth" to "the output" as a whole.

**Circular Logic in Methodology and Results:** The manuscript's conclusion that SMAP SSS closely aligns with GOCI-derived SSS is tautological. Given that GOCI-derived SSS from Kim et al. (2021) was established using SMAP SSS data as "ground truth," it is

logically unsound to use it as a benchmark for validation, leading to an inherent circularity in the comparative analysis.

➔ This study aims to fill the gap in the GOCI-derived SSS data generated by the existing research by Kim et al. (2021). Of course, we are also concerned about the "inherent circularity" you mentioned. Since the GOCI-derived SSS product was generated by the trained model with SMAP SSS as the output, we also used other SSS data (CMEMS and HYCOM) as the input when training the model to avoid the use of the SMAP data, which has been most commonly used. Fortunately, since the validation accuracy of the model using other data showed quality comparable to that of the model using SMAP data, there was not much difference. Thus, we judged that our model using SMAP deviated from the "inherent circulation" issue to some extent.

**Presentation and Organization.** Despite acknowledging the challenges faced by non-native English speakers, the manuscript's organization and language clarity fall below acceptable scientific standards. The presentation and organization of results, particularly in the manuscript's final section, are areas of concern. The narrative primarily enumerates outcomes from various models without effectively synthesizing these findings into coherent conclusions. This approach leaves readers struggling to discern the central thesis and implications of the research. It is critical for the authors to convincingly demonstrate the superiority and applicability of these products for future research with clear evidence. The manuscript currently falls short in this regard, relying on the reader's inference rather than providing direct, substantiated arguments.

➔ Thank you for the comment. We presented the results and discussions together for each result section, and there seemed to be unclear expressions in the process. According to the comment, we revised the sentences to present more clear and comprehensive results based on the research results for each result section.

➔ The following sentences were added in the "4.1 Comparison of the existing SSS with GOCI-derived SSS" Section.

➔ [line 286-288] "Therefore, although reanalysis SSS data is gap-free and has a spatial resolution of about 8 km, it is unsuitable for catching daily SSS spatial fluctuations in the waters because it has relatively low accuracy in waters with low SSS range."

➔ [line 297-300] "Nevertheless, in ECS, which has the characteristics of low SSS during summer, the SMAP SSS data is relatively more consistent with the GOCI-derived SSS than the HYCOM and CMEMS data, so it is more suitable for gap-filling of the GOCI SSS data."

➔ In the "4.2.1 Quantitative evaluation with test dataset", we presented the following sentence to enhance the new findings. [line 324-328] "Notably, the RMSE values of Models 2, 3, and 6 with the bagged trees model and without the SMAP SSS as input showed relatively reasonable levels compared to the models that used for the SMAP SSS as input. This indicates that the bagged trees model overcomes the inconsistencies of the CMEMS and HYCOM SSS concerning the GOCI-derived SSS compared to the fine trees and boosted trees models."

Additionally, it is particularly troublesome that crucial terms are consistently misspelled (e.g., "fine trees" as "find trees", even in the abstract), indicating a lack of thorough proofreading.

While I respect the authors' efforts, the manuscript's frequent errors, especially in critical sections like the abstract, suggest a lack of thorough preparation. This oversight implies a disregard for the peer review process and makes me feel very disappointed.

➔ Thank you for the comment. As a result of our confirmation of the manuscript, the "find trees" case was originally written as "fine trees", but it seems that we haven't reviewed the changes in the proofreading process at the end. We found and revised this part [line 23, 212, 319]. Also, we reviewed the manuscript closely as a whole and revised it.

**Journal Scope Alignment.** ESSD is a journal dedicated to "the publication of articles on original research data (sets), furthering the reuse of high-quality data of benefit to Earth system sciences". The primary emphasis of this manuscript on methodological comparison overshadows the utility and novelty of the data product itself, rendering it better suited for specialized remote sensing journals, such as IEEE Transactions on Geoscience and Remote Sensing, Remote Sensing, or Remote Sensing of Environment. Even for these journals, the first subsection comparing different data products would be more appropriate as supplementary information.

➔ Our study's final daily SSS data is a sea surface salinity map of the East China Sea during the summer with high spatial (0.01°) and temporal resolutions (daily) that have never been seen before. Therefore, we believe that our data are suitable for high-quality data useful for conducting Earth system research and that our data will allow for a variety of previously unavailable studies. Before presenting the accuracy of the models, we checked the data agreement between the input and output data used in the model, which was inserted at the beginning of the result, as we determined that it provided information on the quality of data produced through the model.

In addition, the manuscript claims to present two final SSS products, one from SMAP (Model 1) and the other from GLORYS (Model 2). However, I checked the final output data via the link provided for data download (on ESSD website, the link in the manuscript is non-functional), only one SSS product is accessible. This discrepancy between the stated contributions and the actual accessible data undermines the research's completeness and poses significant concerns regarding data accessibility and transparency.

➔ Thank you for your comment. Through quantitative and qualitative evaluation of the models, we selected Model 1 (only the SMAP SSS as input) and Model 2 (only the CMEMS SSS as input) for the CDW front analysis in the ECS, while considering the simplicity of the input data. Then, when analyzing the CDW front, Model 1 was used from 2015 to 2018, and Model 2 was used for 2019 due to the safe mode of SMAP SSS data. As the final data product, it was combined in this way, and then we uploaded it to the link presented in data availability. I think we lacked a detailed explanation of the data. According to the comments, we divided and uploaded the data of Models 1 and 2.

In summary, while the topic under investigation holds potential for advancing regional oceanographic research, the aforementioned concerns regarding methodology validation, manuscript quality, and alignment with journal scope are too substantive to overlook. A thorough revision addressing these fundamental issues is essential before reconsideration.

Despite the concerns highlighted, the achievement of an R^2 greater than 0.6 across various models is noteworthy and demonstrates the potential value of this study. From the perspective of a possible data user, I offer several suggestions aimed at enhancing the work's reliability and utility. These opinions are primarily based on two considerations: as a data user, how can I trust that this dataset is reliable? What details do I need from the author to utilize the data effectively? I highly recommend considering these improvements and look forward to the potential resubmission of this work.

➔ Thank you for your helpful comments and encouragement to resubmit our work.

**General suggestions:**

**Experiment Design and Data Utilization:** In the development of remote sensing data products, two primary methodologies are commonly employed: (a) When observational samples (Y_obs) are scarce, an algorithm is designed to estimate values (Y_est) independently of these observations. The estimated values are then compared with the observed values (Y_obs) to evaluate accuracy. (b) With a large dataset of observations (Y_obs), the data is divided into two distinct subsets (Y1_obs and Y2_obs), collected at different times or during different cruises. The model is initially trained and tested with Y1_obs, followed by an independent evaluation using Y2_obs. Both strategies offer a convincing foundation for trusting the derived estimates (Y_est) in scenarios where direct observations are unavailable.

The authors opted for the latter strategy. However, the use of GOCI-derived SSS as "ground-truth" or observational data is problematic. I recommend that the authors consider the first approach in writing the manuscript, which does not require that the GOCI-derived SSS are "true values" or not. The significant contribution of this work should be the development of an algorithm capable of producing a SSS data product without relying on direct observations, validated independently through comparisons with NIFS data and (h–n) I-ORS data to ensure the product's reliability.

➔ As the reviewer pointed out, we agree that Kim's GOCI-derived SSS data was described as ground-truth, confusing reviewers. We originally intended the former strategy that the reviewer presented the model design. We trained the model by designating the GOCI-derived SSS data as Y_est, and then evaluated the accuracy with Y_obs (NIFS and I-ORS data).

**Data Product Accessibility: provide a final data product.** It is crucial to provide a clear and easily accessible final data product, instead of multiple choice from multiple models. Enhancing the accessibility of the derived SSS product would significantly improve the manuscript's utility to both readers and potential data users, making it a more practical resource in the field.

➔ To avoid confusion about the choice, we provided the data by Models 1 and 2, respectively. In addition, a description of the data was added to the "5 Data availability" section.

➔ [Line 500-501] "When analyzing the CDW front, Model 1 was used from 2015 to 2018, and Model 2 was used for 2019 due to the safe mode of SMAP SSS data. We provided the SSS dataset of Models 1 and 2 from 2015–2019."

**Uncertainty Analysis.** Offering an explicit estimation of the uncertainties associated with the final data product is essential for end-users. The authors should estimate uncertainties following methodologies from prior studies such as Wang et al., (2014) or Landschützer et al., (2014), thereby enhancing the data's reliability and user trust.

➔ We reviewed the previous studies recommended by the reviewer. We judged that uncertainty analysis was unlikely to be effective in our study due to the spatial and temporal constraints of the field-measured data. Instead, the "4.2 Performance of the SSS models" section was reorganized into three subsections to emphasize the reliability of the data. First, the models were quantitatively evaluated with the test dataset. Second, the models were validated with independent observations. Third, we performed qualitative evaluation using time series analysis and spatial distribution.

**Revisiting the Comparison Framework.** The comparison between SMAP and GOCI-derived SSS may be more effectively presented after the final data product has been robustly defined and derived. A subsequent comparison of this final product with existing data products (SMAP, GLORYS, HYCOM) against independent observational data (from NIFS and I-ORS) would more convincingly demonstrate its advantages, establishing it as the current "best solution".

➔ Thank you for the comment. According to the comment, we added a comparison of existing data with independent observations in the "Results 2.2 Validation with independent observations" section [Table 5, line 385].

**Refinement of the Last Subsection.** Simplify the discussion by focusing on the derived product's improved spatial or temporal resolution and its implications for studying environmental phenomena at new scales. Illustrating specific examples or case studies where the enhanced resolution provides novel insights would underscore the significance of this work in advancing our understanding of Earth system processes. Focus on showcasing advancements that were previously unattainable due to limitations in temporal and spatial precision. Make sure to clearly state these novel contributions in the opening or concluding sentence of the paragraph, ensuring that the reader immediately grasps the significance of the work's higher accuracy in both dimensions.

➔ Thank you for the comment. According to the comment, we added the sentences in "6 Summary" section as follows.

➔ [Line 512-515] "Eventually, the data produced from our study enabled the recognition of SSS distribution and movement patterns of the CDW front in the ECS daily during summer, which were not previously attempted due to spatial and temporal resolution limitations. These results will further advance our understanding and monitoring of long-term SSS variations in the ECS."

**Ref:**

Landschützer, P., Gruber, N., Bakker, D. C. E., & Schuster, U. (2014). Recent variability of the global ocean carbon sink. Global Biogeochemical Cycles, 28(9), 927–949. https://doi.org/10.1002/2014GB004853

Wang, G., Dai, M., Shen, S. S. P., Bai, Y., & Xu, Y. (2014). Quantifying uncertainty sources in the gridded data of sea surface CO 2 partial pressure. Journal of Geophysical Research: Oceans, 119(8), 5181–5189. https://doi.org/10.1002/2013JC009577

**Specific points:**
**Figures and tables:**
1. Figure 2: "Ensemble classifier" in the bottom right corner of image should be "Ensemble regression". Random Forest could be a classifier or regression learner, in this work it is obvious a regression learner.
➔ According to the comment, we revised Figure 2. [line 190]

2. Figure 3: The colorbars for panels g to i are unclear. If these are 2-D density plots, the bin interval should be introduced.
➔ We added the color bars in each panel of Figure 3g-i. [line 305]

**Main text**
**Introduction**
3. Line 40: "Because waters affected by river outflow and coastal regions are characterized by short- term variability, gridded SSS products can provide useful information for monitoring SSS variations".
This statement needs clarification. While gridded SSS products can indeed provide useful information for monitoring SSS variations, this capability is not necessarily due to the presence of river outflows or short-term variability in coastal regions.
➔ To avoid confusion, we revised it as follows: [line 40-42] "In particular, gridded SSS products can provide useful information for monitoring SSS variations in waters affected by river outflow and coastal regions".

4. Line 57-59: "First, the accuracy of in situ observations defines how information is propagated from data-rich to data-sparse regions and is critically dependent on data coverage and the reliability of spatial covariance".
This statement needs revision. The abbreviated version of this sentence: "The accuracy of observations defined information propagation," and "The accuracy of observations depends on data coverage and spatial covariance"
The accuracy of observations cannot define these. The author likely intends to convey that in regions with low observational coverage, the available data may not accurately represent the phenomena of interest.
➔ According to the comment, we revised it as follows: [line 57-59] "First, in situ observations are characterized by temporal and spatial constraints, and in situ observation accuracy is susceptible to influence by data ranges and regions."

Data Section:
5. Line 125: "The SMAP, HYCOM … were used …, HYCOM is a … model …. We used GOFS Global analysis data".
These sentences need to clarify the relationship between HYCOM and GOFS. It would be beneficial to reorganize the sentences to explicitly state the connection, such as: "HYCOM is a ... model, which forms the computational core of GOFS," or "HYCOM is a ... model within the GOFS."
➔ According to the comment, we revised it as follows: [line 121-122] "HYCOM is a ... model, which forms the computational core of Global Ocean Forecasting System (GOFS)."

6. Line 129: "The GLORYS12V1 product is the CMEMS global ocean eddy-resolving reanalysis covering altimetry at 0.08° × 0.08° and 50 standard levels.".
However, the phrase "covering altimetry" may benefit from clarification. It is likely that "covering altimetry" implies either that the GLORYS12V1 product assimilates altimetry

data into its reanalysis or that the GLORYS12V1 reanalysis spans the altimetry era, which began in 1993.

→ According to the comment, we revised it as follows: [line 127-128] "The GLORYS12V1 product is the CMEMS global ocean eddy-resolving reanalysis and assimilates altimetry data. It has a spatial resolution of 0.08° × 0.08° and 50 standard levels."

7. Line 136: "at the highest spatial resolution",
The expression "highest" is inappropriate. It is recommended to change it to "at a high spatial resolution."

→ According to the comment, we revised it as follows: [line 134] "at a high spatial resolution"

8. Line 140: There is no introduction about the temporal resolution of ERA5. I guessed that the temporal resolution should be hourly.

→ According to the comment, we revised it as follows: [line 138] "The data frequency is hourly and daily mean data was used."

9. Line 153: "SSS data obtained from the ESC, West sea, and South sea".
The locations of "West sea and South sea" are unclear in Fig.1. Are the authors referring to the western and southern marginal seas of the Korean peninsula?

→ To avoid confusion, we added "West Sea" and "South Sea". [Figure 1, line 164]

10. Section 2.3. In situ data: An introduction regarding the uncertainty of the salinity measurements should be provided.

→ The National Institute of Fisheries Science (NIFS) and the Korea Institute of Ocean Science and Technology (KIOST) give a QC flag to the observational data based on the recommended standards for marine data by the UNESCO Intergovernmental Oceanographic Commission (IOC): QC Flag 1 (Good), 2 (not evaluated, not available or unknown), 3 (Questionable/suspect), 4 (Bad), and 9 (Missing data).

→ To secure reliable data, we used the data with QC flag 1 (good). We added the following sentence. [line 157-158] "All data were used as the quality control (QC) flag 1 (good), and the specified measurement accuracy is $\pm 0.003$ psu."

Method section

11. Line 172: "All data were sampled at 0.01°",
I guessed that the SSS, SSH, uo, vo, and wind stress curl were all interpolated into 0.01° resolution. Clarification on this point is requested.

→ According to the comment, we revised it as follows: [line 172-173] "To match the spatial resolution of the gridded maps, input and output data as shown in Table 1 were sampled at 0.01°, which is the spatial resolution of the SST level."

12. Line 181: "This method is more effective than other methods when the data vary rapidly",
This statement lacks objectivity, as it claims superiority over unspecified "other methods" without providing citations or clarifying the basis for comparison.

→ To avoid confusion, we revised it as follows: [line 183-184] "This method is more effective when the data vary rapidly."

13. Line 202: "the input groups … exhibited 500,000 matched pixel pairs or more",

It might be better to simply use "had" instead of "exhibited" for clarity.

➔ According to the comment, we revised it as follows: [line 202-203] "the input groups … had 500,000 matched pixel pairs or more"

14. Line 203: "We then added as many as 10% of the matched pixel pairs to each input group.".
This sentence is unclear. The phrase "as many as" is unnecessary. Based on the following text, it seems the author intended to say, "For each training and test set, we then added 10% of its total number of zero matrices." Additionally, this approach is confusing, as the purpose and benefits are not introduced. A literature reference or justification for this method would be helpful, as it potentially reduces the signal-to-noise ratio.

➔ According to the comment, we revised it as follows: [line 204-209] Due to differences in spatial resolution between input data, the location of non-valued pixels differs by input data, so they were adopted as pixel pairs only if all the input values were available. This is because the accuracy of the trained model is degraded if non-valued pixels are included in input dataset. Then, if at least one of the input data had non-valued pixels, all values of the pixel pairs were converted to zero values. For the training of zero values within the matched images, we added 10% of the total number of zero matrices for each training and test dataset group."

15. Line 208: "find trees".
The word "find" should be "fine". The same mistake is also made in line 22 of the Abstract, Line 317 of the Section 4.2.1. Quantitative evaluation.

➔ According to the comment, we revised this part [line 23, 212, 319].

16. Line 215 to 219: These sentences are not necessary.
These sentences are not necessary, as the preceding sentences have already introduced the decision tree and two random forest ensemble algorithms. The reader can infer that the computational time will rank as follows: bagged trees > boosted trees > decision tree.

➔ According to the comment, we revised it as follows: [line 219-220] "The computational times rank as follows: Bagged trees, boosted trees, and fine trees."

17. Line 214. There is a typo; "booted trees" should be "boosted trees". To be honest I'm a bit disappointed, the authors shouldn't have made such a basic mistake in such important parts of the article.

➔ Thank you for the comment. We revised the word [line 219]. We corrected the typo as a whole.

18. Section 3.2. It is confusing to use x to represent the GOCI-derived SSS (the "ground-truth" values) and y to represent the "compared SSS." It would be preferable to use y_tru to represent "ground-truth values" and y_est to represent "estimated values.".

➔ To avoid confusion, we revised it as follows: [line 239-240] "x represents the GOCI-derived SSS or in situ observation SSS. y represents the SSS products and the estimated SSS."

19. Equation (1) is the squared correlation coefficient, I prefer to use the equation: $R^2 = 1 - [\Sigma(y_i - \hat{y}_i)^2 / \Sigma(y_i - \bar{y})^2]$, as it directly represents the coefficient of determination, rather than being equivalent to the squared correlation coefficient. The authors can choose to deny my opinion on this point since they have the same values.

➔ Since the equation is the same as that suggested by the reviewer, we intend to use this equation.

Results and discussions

20. Line 241: "distribution trends" is somewhat unusual and could be benefit from clarification or rephrasing. According to the following sentences, I think it should be "…, we examined the spatial and statistical distribution of xxx data products …"

➔ According to the comment, we revised it as follows: [line 243-244] "we examined the statistical distribution of the SMAP, CMEMS, and HYCOM SSS products…"

21. Line 258: "it did not reflect the daily SSS product because it was an 8-days average product",
The expression "reflect the daily product" is strange. It is guessed that the authors intended to convey that the SMAP dataset does not have a daily resolution.

➔ According to the comment, we revised it as follows: [line 260] "however, it is an 8-day average product, not a daily product."

22. Line 296: "However, the SMAP SSS in our study area showed a more reasonable degree of agreement with the in situ NIFS SSS compared to that of the reanalysis SSS; hence, the SMAP SSS data can be a good reference for monitoring the CDW in the ECS.".
There are no figures or statistics provided in this manuscript to support this conclusion. Section 4.1 and Figure 3 are comparing three SSS data products with the GOCI-derived SSS.

➔ To avoid the confusion, we removed this sentence.

**4.2 Performance of the SSS models**

23. Line 336: "Because to the difference in spatial resolution among the data, the pixels masked at zero were different;",
The expression "masked at zero" is misleading. I think the authors intended to convey that those pixels with missing values were different among the datasets. However, using "masked at zero" is misleading because temperature, salinity, and other variables have meaning when they equal zero.

24. Line 338: "The model was trained using zero-free data, thereby not properly recognizing the actual mask pixels, resulting in a specific value of pixels in the masked area.". The meaning of "specific value" in this context is unclear.

25. Line 340: "we added as many as 10% of the matched pixel pairs for each model",
Again, clarification is needed regarding these 10% pixel pairs. Are they repeated samples using the bootstrap method or zero values?

➔ Regarding Q23, Q24, and Q25, we removed these sentences and described their contents in "3.1 machine learning models" Section instead.

➔ [line 204-209] "Due to differences in spatial resolution between input data, the location of non-valued pixels differs by input data, so they were adopted as pixel pairs only if all the input values were available. This is because the accuracy of the trained model is degraded if non-valued pixels are included in input dataset. Then, if at least one of the input data had non-valued pixels, all values of the pixel pairs were converted to zero values. For the training of zero values within the matched images, we added 10% of the total number of zero matrices for each training and test dataset group."

26. Line 381-385: Please provide the specific RMSE values. A 5.36% increase or 12.32% decrease may be small depending on the absolute value. For instance, assuming an RMSE of 1.5, those changes would only be 1.58 and 1.32, respectively.

➔ According to the comment, we revised it as follows: [line 373-375] "When using in situ data in the >31 psu range, the mean RMSE (1.573 psu) was 5.36% higher than the mean RMSE the entire dataset (1.493 psu), and the performances of all models were slightly worse. However, when in situ data in the <31 psu range were used, the mean RMSE (1.308 psu) decreased by 12.32% compared to the mean RMSE of the entire dataset."

27. Line 393: "The RMSE of Model 5, which had the worst performance among the models.", Grammatically, this is an incomplete sentence. It requires further elaboration.

➔ We removed this sentence.

28. Line 394 – 396: "As shown in table 3, the three SSS datasets showed high accuracies in the >31 psu range; therefore, using the in situ NIFS dataset resulted in low RMSE values. However, the in situ I-ORS data have a high RMSE because most of the SSS data are in the <31 psu range."
NIFS and I-ORS are both observational datasets and, therefore, cannot have RMSE values themselves. It is the comparison between these observations and the three data products that yields the RMSE.

29. Line 398: "In summary, all bagged trees models trained with a combination of various input variables could estimate the SSS of the CDW in the ECS on a daily basis with RMSE values of less than 1 psu, i.e., higher than that of the SMAP SSS."
There are two issues with this statement:
a. The RMSE values of comparisons between NIFS and the models are larger than 1.0. Only the comparisons between most model-derived SSS and I-ORS have RMSE < 1 (i.e., Fig. 4h-n). Therefore, it should state "most bagged trees models" instead of "all models," as Figure 4j and 4l have RMSE = 1.0 and = 1.02, respectively.
b. The phrase "higher than that of the SMAP SSS" should be replaced with "lower" or "better" to accurately convey the intended meaning.

➔ Regarding Q 28 and Q9, we removed the original sentences and added the new sentence as follows: [379-380] "This indicates that in a water environment with a low salinity range, the SSS data estimated by our models have a higher accuracy than the existing SSS products, approximately the accuracy level of RMSE = 1 psu."

**Data availability**
30. The link doesn't work, and I go to the preprint dataset page, it should be https://doi.org/10.22808/DATA-2023-2.

➔ We revised the link to https://doi.org/10.22808/DATA-2023-2. [line 499]

**Small typo:**
31. Line 64: "There are only few in situ SSS observations …",
"a few" or "few"? "few" means there are no in ARGO floats at all.

➔ We revised it as follows: [line 63-64] "there are a few in situ SSS observations…"

32. Line 105: "because to" should be "because of"

➔ We revised it as follows: [line 104-105] "because of frequent cloud cover, sun glint, and thick aerosols."

33. Line 180: "We applied to", remove "to"
   ➜ We revised it as follows: [line 181] "we applied to a Savitzky–Golay filter…"

34. Line 253: ".. all pixels in the study area cloud not provide SSS information", It should be "could not" but "cloud not".
   ➜ We revised it as follows: [line 255] "all pixels in the study area could not provide …"

35. Line 286: "Currently, the SMAP are the only satellite data", I think here "are" should be "is" because the subject SMAP is singular.
   ➜ We revised it as follows: [line 288] "the SMAP is the only satellite data…"

36. Line 431: There is a typo; "patter" should be "pattern."
   ➜ We revised it as follows: [line 414] "showed the appropriate CDW distributions and patterns.…"

37. Line 449: "We identified the three phases according the CDW", "according" should be "according to"
   ➜ We revised it as follows: [line 434] "according to the CDW variations during summer:.…"

---

## Author Comment (AC2)

**[RC2]**

Review of Gap-filling processes on GOCI-derived daily sea surface salinity product for Changjiang diluted water front in the East China Sea by Jisun Shin, Dae-Won Kim, So-Hyun Kim, Gi Seop Lee, Boo-Keun Kim, Young-Heon Jo.

The paper presents a method to generate daily gap-filled sea surface salinity fields using as input lower resolution passive microwave data taking into account the correlation between ocean color and salinity fields and exploiting machine learning. The study focuses on the short term evolution of the Changjiang diluted water (CDW) front. It is essentially ok for publication after the recommendation below are followed.

My first recommendation is to modify the title that in its current form suggests that there are gap-filling processes at work in the East China Sea while I believe the authors should use instead of "Gap-filling processes on GOCI-derived…" the following: "Gap-filling techniques applied to GOCI-derived…" It would important if the difference in methodology between *Kim, D. W., Kim, S. H., and Jo, Y. H.: Machine Learning to Identify Three Types of Oceanic Fronts Associated with the Changjiang Diluted Water in the East China Sea between 1997 and 2021, Remote Sens., 14(15), 3574, doi:10.3390/rs14153574, 2022b* and the current paper were clearly explained to show the reader the novelty of the current approach.

➔ Thank you for the comment. We agree that it is important to show the novelty of our study. According to the comment, we revised the title.

➔ "Gap-filling techniques applied to GOCI-derived daily sea surface salinity product for Changjiang diluted water front in the East China Sea"

After the first definition of GOCI (Line 15) do not spell it out in the text and figure legends.

➔ We confirmed and revised this part.

➔ [Line 165, Table 1 caption]: "Table 1. Summary of inputs and output used for training and testing of the machine learning model. The output data was used for daily SSS map derived from Geostationary Ocean Color Imager (GOCI) by Kim et al. (2021). In situ SSS data for the model testing were provided by the NIFS and I-ORS."

The following figures needs to be regenerated with increased resolution: 2, 3, 4, 5, 7 and 8

➔ We increased the resolution of the figures and re-inserted it into the word file. The resolution of the original figures inserted in the word file is high, but it seems that the resolution decreased during the preprint process.

Detailed suggested changes to improve readability and correct apparent mistakes:
Line 13 Replace "high" with "higher"

➔ According to the comment, we revised the word in the manuscript [line 14].

Line 16 Replace "season from with "seasons of"

➔ We revised this part as follows: [line 17-18] "during the summer seasons from 2015 to 2019".

Line 17 Replace "Copernicus Marine Environment Monitoring Service" with "Copernicus Marine Service"

➔ We referred to these full names in the following sites.

➔ https://insitu.copernicus.eu/FactSheets/CMEMS/

Line 30 Replace "horizontal" with "zonal"
➔ According to the comment, we revised the word in the manuscript [line 31].

Line 37 remove "—the salinity of the ocean at its surface—"
➔ According to the comment, we removed the part in the manuscript [line 38].

Line 38 replace "dataset" with "products"
➔ According to the comment, we revised the word in the manuscript [line 39].

Line 69 Replace "Copernicus Marine Environment Monitoring Service" with "Copernicus Marine Service"
➔ We referred to these full names in the following sites.
➔ "https://insitu.copernicus.eu/FactSheets/CMEMS/"

Line 130 replace and track with "along track"
➔ According to the comment, we revised the word in the manuscript [line 128].

Line 130-131 replace "using a reduced-order Kalman filter, and track altimeter data, satellite SST and in situ TS profiles were jointly assimilated" with "using a reduced-order Kalman filter, along track altimeter data, satellite SST, and in situ TS profiles."
➔ According to the comment, we revised the word in the manuscript [line 128-129].

Line 184 Replace "horizontal" with "zonal"
➔ According to the comment, we revised the word in the manuscript [line 182].

Line 431    Replace "patter" with "patterns"
➔ According to the comment, we revised the word in the manuscript [line 414].

Line 518 & 519 It seems from the manuscript that ">31 psu" (at line 518) should be swapped with "<31 psu" (at line 519)
➔ ">31 psu" means 31 psu or more, so this sentence is correct.

Line 519 What is meant by "most oceanic environment"? "Typical" perhaps?
➔ To avoid confusion, we revised as follows: [line 505] "the <31 psu range in the ECS during summer."

In Table 4: Model 1 (SMAP) with bagged trees reports the best RMSE 1.17 psu with 1.36 MSE however, we have either 1.36 MSE with 1.17 RMSE or 1.35 MSE with 1.16 RMSE. So a rounding off error was made. Since this model was selected as the best, from validation results, it is a relevant issue.
➔ To avoid rounding errors, we expressed the values of RMSE, MSE, MAE in Table 4 up to three decimal places.